# Classroom effects of a preventive behavioral management program: A pragmatic cluster-randomized trial of Good Behavior Game

Dariush Djamnezhad [1,2*], Martin Bergström [3], Carl Delfin [4,5], Björn Hofvander [1,5,6]

1 Lund Clinical Research on Externalizing and Developmental Psychopathology, Department of Clinical Sciences Lund, Lund University, Lund, Sweden, 2 Administration of Compulsory Education Department, City of Malmö, Malmö, Sweden, 3 School of Social Work, Lund University, Lund, Sweden, 4 Evidence-based Forensic Psychiatry, Department of Clinical Sciences Lund, Lund University, Lund, Sweden, 5 Centre of Ethics, Law and Mental Health, Department of Psychiatry and Neurochemistry, University of Gothenburg, Gothenburg, Sweden, 6 Department of Forensic Psychiatry, Skåne University Hospital, Trelleborg, Sweden

* dariush.djamnezhad@med.lu.se

## Abstract

Good Behavior Game (GBG) is a school-based intervention designed to reduce conduct problems, while increasing on-task behavior and a positive classroom climate. Earlier studies have shown positive long-term effects in several outcomes, making GBG a promising method of universal prevention. This study evaluates the effectiveness of a translated and adapted version of GBG in a pragmatic, cluster-randomized controlled, parallel group superiority trial. Schools with K–3 students were eligible for recruitment. Five schools were recruited to either receive training in GBG or continue with business-as-usual. Schools were allocated using stratified randomization. The outcomes included teacher-rated conduct problems in the classroom (primary outcome) and common school areas, observer-rated on-task behavior, along with teacher- and observer-rated classroom climate. All 43 classrooms had teacher-rated measures, while a subset of 20 classrooms were randomly assigned to also receive the observer-rated measures by blinded and independent observers. Measurements were conducted at the start (pre-intervention), middle (3-month follow-up), and end of the school year (9-month follow-up and primary endpoint). Measurements had a 100% response rate and all classrooms were included in the analysis using Bayesian mixed effects models. Conduct problems in the classroom, on-task behavior and classroom climate had effects in a positive direction. Partially in line with the mechanics of GBG, there was no effect on conduct problems in common school areas over time. Effects for all outcomes were imprecise, as highest posterior density intervals overlapped for changes over time between the intervention and control group. Although imprecise, effects were directionally consistent with earlier research. Taken together, this study lends tentative support for cross-cultural transportation

**Data availability statement:** All data (with associated metadata) underlying the findings, along with statistical code used to prepare and analyze the data, is publicly available at Researchdata.se (https://doi.org/10.5878/s9bv-0w05).

**Funding:** This study is funded by the City of Malmö (DD), Jerringfonden (BH), Majblomman (BH), Jane and Dan Olsson Foundations (BH), Fredrik och Ingrid Thurings Stiftelse (DD; registration number: 2024-100), and Stiftelsen Clas Groschinskys minnesfond (DD; registration number: SF2517). The funders had no role in study design, data collection and analysis, decision to publish, or preparation of the manuscript.

**Competing interests:** I have read the journal's policy and the authors of this manuscript have the following competing interests: DD is currently a member of the same not-for-profit organizations (City of Malmö and Prevention Sverige) as several CTC-facilitators and GBG-trainers. The other authors have no other interests to declare.

and adaptation of GBG to a pragmatic context, without major external provision of resources or personnel. Trial registration: ClinicalTrials.gov, No. NCT05794893.

## Introduction

Universal school-based programs targeting children's well-being and non-academic skills have seen an increase in recent years [1,2]. From a preventive standpoint the rationale is simple: almost all children spend a large amount of time in classrooms; thus, universal prevention (also known as primary prevention or tier 1) will reach early and wide. Although educational settings usually have academic goals in mind, universal school-based interventions can be designed to have children learn important social and emotional skills beyond what they otherwise would with only academic instruction [3]. While these skills are pivotal in their own right, a central aim in this type of intervention is usually to reduce conduct problems [4,5]. Generally, conduct problems in early childhood is associated with later substance use, poor mental and physical health, poor educational and occupational attainment, and criminal behavior, compared to low levels of conduct problems in childhood [6]. Conduct problems, such as aggressive or disruptive behaviors, are thought to create and reinforce poor or antagonistic relationships between children, their peers, and their teachers. This in turn supports conditions for academic disengagement and contact with deviating peers, which may be followed by risk-taking and antisocial behaviors. Programs that increase social and emotional skills (e.g., self-regulation) and promote a positive classroom climate are recognized as effective interventions for reducing conduct problems in schools [7], which should help disrupt this negative cycle. The related improvements in psychosocial climate, relationships and general learning environment should constitute an important foundation for children's mental health and wellbeing [8].

### Good behavior game

A program often mentioned in the context of universal school-based prevention is Good Behavior Game (GBG), particularly as a program showing reductions in children's conduct problems [9]. Although GBG isn't an entirely uniform intervention (e.g., a widely disseminated version is PAX GBG [10]), some common features are the use of rules, interdependent teams, and rewards [11]. These components are integrated with ordinary classroom lessons and procedures in a game-like fashion to reinforce desired student behavior and discourage rule-breaking [11].

GBG has a long history of research, first appearing in literature more than 50 years ago [12]. Heavily influenced by principles of applied behavior analysis, the first wave of studies focused on single-case designs [13–16]. An important development occurred as the study of GBG expanded to randomized controlled trials, demonstrating both short- [17], and long-term positive effects on a large scale [18–23]. Today GBG has been studied in numerous contexts, including different cultural settings and countries [24–26], as well as student populations of varying age, educational setting, and ability [27–33].

## Challenges

While the literature surrounding GBG is substantial, a key issue is the robustness of effects as the configuration of GBG, the trial, and context varies. There are a multitude of factors connected to this. Transported interventions are often less efficacious [34], although not necessarily a rule, it has been known to happen to both GBG and similar interventions [35,36]. Maintaining high implementation fidelity is often seen as important for positive effects [37], but somewhat contradictory it may also be important to adapt the intervention [38]. While the departure from the original trial context alone warrants additional evaluation [39], just different trial stages and conditions in themselves may significantly impact results of school-based interventions [40]. To evaluate efficacy under optimal conditions, schools are often provided additional resources and personnel during the trial [7], but these circumstances are likely dissimilar to the strained realities of naturalistic conditions where additional support might be crucial for sustaining or scaling up interventions beyond the trial [41–44]. In one example, without explicit support contingent on the trial, the proportion of high-implementing schools went from 70% to 30% as the school-based intervention was scaled-up outside the trial context [45]. These findings advocate the addition of pragmatic trials [46].

## The current study

In accordance with the need for GBG to be evaluated under transported, adapted and naturalistic circumstances, this is the first randomized trial studying the effectiveness of GBG in a Nordic setting. Although Sweden, where this trial takes place, is a high income welfare state, with comparatively high spending on education [47], behavioral issues in schools are not significantly lower compared to other countries [48]. It has been a national concern for more than 10 years [49–51], and considered one of the top priorities for professional development among Swedish teachers [52]. The concern is also shared by surrounding Nordic countries [53,54].

Despite an apparent need, there are no national institutions or similar that provide schools with evidence-based behavioral interventions, and only a few school-based behavioral interventions undergo scientific evaluation in Sweden [55,56]. While there may be some support through national regulations and agencies, the complicated and resource-intensive endeavor of transporting or creating novel school-based interventions is often left up to the local municipality, who retains primary responsibility for public schools in Sweden.

This trial seized the opportunity to study the efforts of a Swedish municipality (i.e., a pragmatic, public sector, non-research organization) that has translated, adapted, and built capacity to implement GBG. As stated in the study protocol [57], this study has explicitly aspired to conduct a trial that is on the far end on a pragmatic spectrum. The trial attempted to affect the usual practices of the municipality as little as possible, allowing the trainers to train teachers in GBG per their usual practice, while having a comparable control group of classrooms that follow usual practice without training in GBG.

## Aims

The objective is to evaluate the effects of teachers receiving GBG-training in elementary school classrooms on outcomes related to student behaviors, as compared to schools conducting business-as-usual (BAU). The outcomes and hypotheses are chosen to be consistent with the theoretical base and logic model of GBG to see if the intervention effects follow the intended mechanisms [58]. The occurrence of conduct problems in the classroom is chosen as the primary outcome based on its importance as a short-term outcome for schools, and as a pivotal mechanism in how GBG is theorized to function as a long-term preventive intervention. We first hypothesize that conduct problems in the classroom will have a greater decrease over time in the GBG-group compared to BAU at both 3- and 9-month follow-up (Hypothesis 1). Secondly, we hypothesize that conduct problems in common school areas will have a greater decrease at the 9-month follow-up compared to BAU, but not at the 3-month follow-up as generalizing activities in GBG are not expected to be implemented that early (Hypothesis 2). Moreover, we hypothesize that classroom climate (both teacher- and

observer-rated) will have a greater increase over time in the GBG-group compared to BAU at the 9-month follow-up (Hypothesis 3). Finally, we hypothesize that on-task behavior will have a greater relative increase over time in the GBG-group compared to BAU at the 9-month follow-up (Hypothesis 4).

## Method

This is a pragmatic, cluster-randomized controlled, parallel group superiority trial and is reported according to CONSORT 2025 [59], with extensions for cluster-randomized trials [60], pragmatic trials [61], and social and psychological interventions [62]. These checklists are combined and provided in S1 CONSORT Checklist. The trial was retrospectively registered at ClinicalTrials.gov (NCT05794893) on April 3, 2023, https://clinicaltrials.gov/study/NCT05794893. Additional details about the full study are available in the study protocol [57].

All schools were simultaneously assigned during randomization, which was conducted and sent to schools on June 2–3, 2021, by the research group. An aggregated sociodemographic variable was used for stratification, on the basis of sociodemographic variables being related to conduct problems [63], and schools were chosen as the unit of randomization.

### Ethics statement

This study was approved by the Swedish Ethical Review Authority (registration number: 2020–06804). The requirement for informed consent by participants and/or their legal guardians was waived as the study does not collect personal or sensitive information.

### Setting, recruitment and sample

This study was conducted in Malmö, Sweden´s third largest municipality, who piloted an established implementation infrastructure and framework for universal prevention, Communities That Care (CTC) [64]. The GBG implementation team (consisting of GBG-trainers) operated out of this context. A steering committee was formed to include representatives from the GBG implementation team, CTC-facilitators and the research group, in order to efficiently coordinate the trial, and to make sure that the trial was practically relevant and feasible.

The study work group included a coordinator (DD) who had contact with designated site-coordinators at each school, GBG-trainers, observers, and the rest of the research group. The primary role of the research group was to complete randomization and ensure fidelity to the data collection process, otherwise affecting as little as possible to ensure that both allocations mirrored usual practice. Site-coordinators were in place to facilitate communication between the research group and schools, primarily regarding data collection. School principals had final say in joining the trial and assigning site coordinators.

The trial followed the municipality's usual practice for recruiting schools to be offered GBG-training. Eligible schools were schools in the municipality with any K–3 (corresponding to children aged 6–9) classrooms. Schools that previously had implemented GBG or primarily served special needs students were not eligible. All K–3 classrooms in included schools were enrolled in the study. This was primarily to follow the municipality's usual practice of implementing GBG on a per-school basis (i.e., all K–3 classrooms in each school are required to participate), while also minimizing contamination effects between proximal classrooms (e.g., GBG-trained teachers spreading core components to colleagues). Note however that GBG itself is primarily implemented on a classroom level, which is the unit of analysis.

The first phase of recruitment consisted of CTC-facilitators and GBG-trainers recruiting schools in pilot areas based on a needs assessment, i.e., whether risk- and protective factors in the area, as measured by the CTC Youth Survey [65], matched the logic model of GBG. This phase ended in April 2021, at which point only three schools had been included. As this was below capacity for the GBG implementation team, recruitment expanded to a second phase

where 40 eligible schools were given the opportunity to join the trial. This phase was primarily headed by the GBG implementation team and ended in May 2021, netting a total of eight schools. Unfortunately, three schools dropped out just prior to randomization, leaving only five schools at the end of the recruitment phase. The research group decided to continue with the study as the municipality was expected to keep training schools in GBG regardless, thereby decreasing the pool of eligible and willing schools each year. The final allocation was then 3:2 (planned allocation ratio was 1:1) with an additional school being randomized to the intervention group, primarily to avoid needlessly delaying the intervention and lowering the capacity of the GBG implementation team. The process is summarized in the corresponding CONSORT flow diagram (Fig 1).

The sample size was largely based on the municipality's capacity for implementing GBG rather than statistical power. Although a pre-specified simulation specific to this study would have been preferable, earlier school-based simulations show relatively low bias when using Bayesian mixed models with priors and fixed effects in relevant

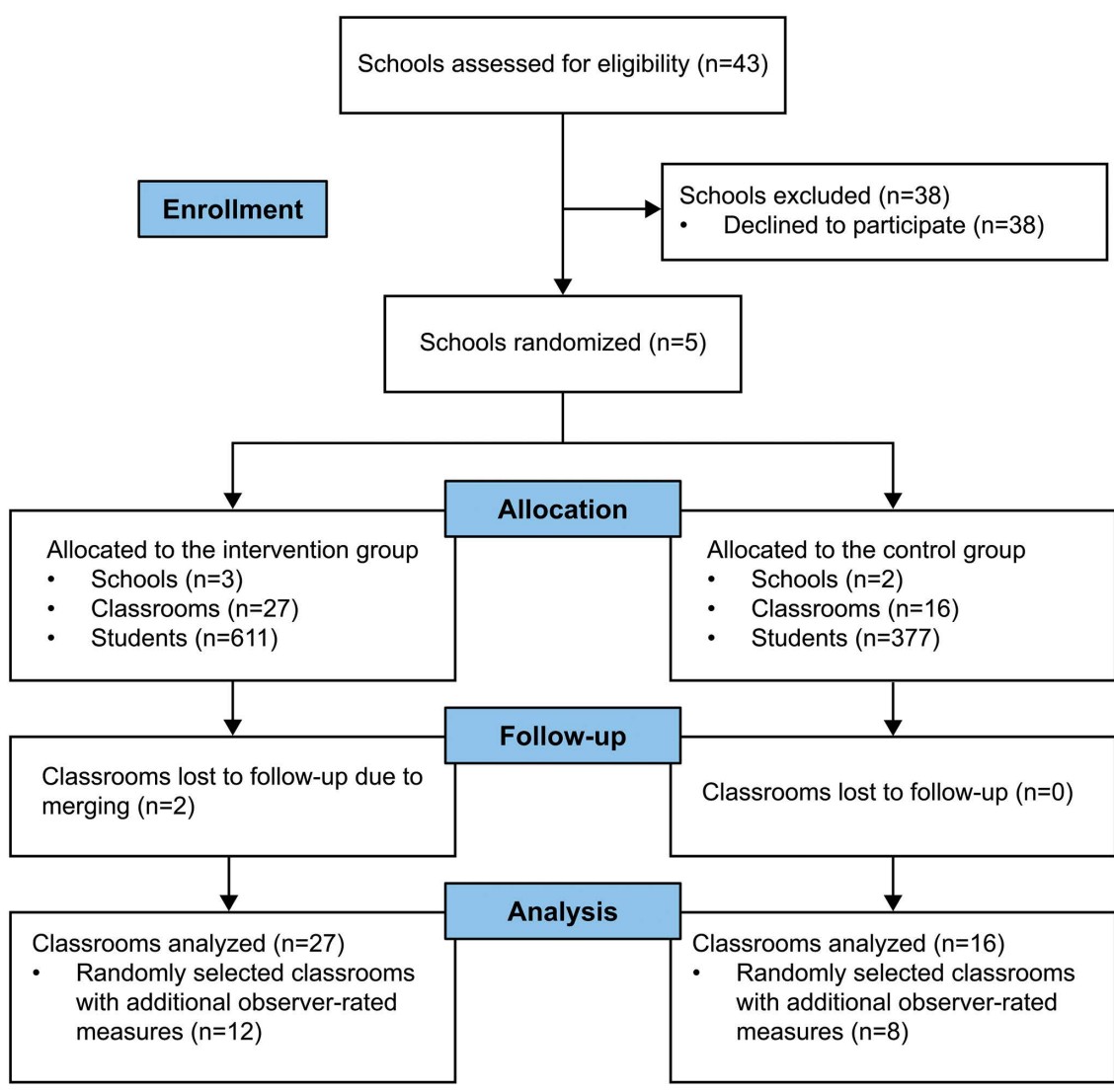

**Fig 1. CONSORT flow diagram.**

estimates, even with very few clusters [66]. While extending GBG-implementation and the trial to other municipalities was considered, it was not part of usual practice, nor was it clear if it was supported by infrastructure and capacity at the time.

The only incentives provided by the study in the recruitment phase was that schools randomized to the BAU-group would be prioritized for the next wave of GBG-training (scheduled for the next school year) and that the research group would report back preliminary local data collected at the specific school. No additional resources were provided to schools during the trial.

## Interventions

Training in GBG was wholly organized and provided by the municipality's implementation team to the intervention group. As previously mentioned, there are variations of GBG in the literature. This version of GBG (named *Höjaspelet*) is a manualized and game-like intervention designed to be integrated into ordinary educational activities by teachers in K–3 classrooms. The game is played using balanced teams, i.e., the students who most frequently break classroom rules are in different teams. During the game, students are asked to follow three positively stated rules for classroom behavior. The teacher will try to give behavior-specific praise to students and teams following the rules, while rule infractions will cause the teacher to discreetly draw point cards from the team. All teams with at least one point card left will have won the game, which earns a tally on the team poster, additional praise, and potential rewards (e.g., a preferred activity or small tangibles).

The first year of GBG, which is the focus of this study, is sequenced into three phases: the introductory phase (September to December), the expansion phase (January to March), and the generalization phase (April to June). Progressing into a new phase will generally consist of creating new teams, increasing time played each session, expanding the game or components to new classroom contexts (e.g., from independent work to whole- or small group activities in the classroom), and fading out the frequency of rewards. The game is played three times a week regardless of phase. Game time usually starts out at 10 minutes but is gradually increased.

Training in GBG consisted of three sessions of group instruction (approximately 3,5h workshops) that preceded each phase, along with nine observations. Workshops were typically conducted at central locations to train teachers from multiple schools simultaneously. Observations were conducted on-site and coupled with written feedback, with the addition of oral feedback if logistically permitted. This was besides any accompanying activities conducted by teachers independently, such as reading the manual or otherwise preparing for game sessions. Trainers could also have additional contact with intervention schools and their teachers regarding implementation, e.g., organizational matters or technical support. Training in GBG after the first year (post-certification) is usually continued by booster sessions and training on-site teachers to be in-house coaches.

The manual and materials are not publicly or otherwise widely available at this point. They are provided by the municipality's implementation team to each classroom when commencing implementation. Interested parties are referred to the City of Malmö for further information. Most of the GBG implementation team at the time of study had backgrounds as teachers, with one trainer having a background as a school counselor. The municipality's implementation team primarily used two GBG-trainers to train the three schools in the intervention group, though other trainers were present at the workshops. One of the primary trainers was first-generation and externally certified. The other trainer was second-generation and certified in-house.

The control group followed BAU, such that schools and teachers were free to use all practices available to them without restriction. The only limitation was that the control group couldn't receive direct training in GBG from the municipality's GBG-trainers. The research group made no attempts to support or control practice in the intervention- or control group, save for ensuring fidelity to the data collection process. Teacher practices were partially tracked in both groups, which will be further investigated in an upcoming paper.

## Outcomes

This study used teacher- and observer-rated measures. All measures were on the classroom level. Data collection was concentrated at three time points, with each data collection period being constricted to a maximum of three weeks. The length of the study and intervention is aligned to a typical Swedish school year. T1 (baseline) was in August/September 2021 before start of GBG-training. T2 (3-month follow-up) was in November/December 2021 before winter break and second phase of GBG, T3 (9-month follow-up, primary endpoint) was in May/June 2022 before end of the school year. The exact dates were data collection officially started and ended were August 17, 2021, and June 14, 2022, respectively. No systematic accounting of harms outside available outcomes was conducted.

Where applicable, internal consistency is reported for outcomes using multilevel composite reliability, $\omega_{within}$ and $\omega_{between}$ [67], estimated with the R-package multilevelTools [68].

## Teacher ratings

All classrooms provided teacher-rated measures. In most cases, the primary teacher for the class filled out a teacher form containing the teacher-rated outcomes and demographic items at a given time-point.

Teachers provided demographic information at T1 and T3 which included the number of students in their class, how many of them were boys, how many of them had Swedish as a secondary language, how many students that were wholly or partially excluded from ordinary classroom activities for at least 3 lessons per week (e.g., for special education purposes), whether they had additional staff, and how much experience the teacher had with the current class.

## Observer ratings

In each school, one classroom per grade level also received additional measures rated by blinded and independent observers. If a school had multiple classrooms in a grade level, the classroom receiving observer-ratings was randomly chosen prior to baseline.

Seven independent observers were hired by the research group, primarily recruited from graduate- and undergraduate programs in pedagogy, social work, or psychology. The observers were instructed during a full-day workshop where they also received a detailed manual and observational forms. The workshop included instructions regarding observational measures, reliability training, and data collection logistics. The workshop was facilitated using recorded video material from a K–3 classroom. Observers were randomized to schools, following the same set of classrooms T1–T3, and conducted each observation solo.

The observers were blinded to study design and intervention. They were given a general description that the study was about behavioral management and the classroom environment, and that further information will be given after the completion of data collection. Schools were asked not to discuss anything pertaining to GBG or study design with the observers. Observations were booked independently between observers and schools. Schools in the intervention group were asked to not have observers present during GBG-sessions, as this study is more focused on how outcomes generalize outside game sessions. There was no systematic assessment to see if blinding had succeeded. There was no other active blinding of participants, intervention providers, or raters in the study, as it was deemed too difficult to achieve systematically within the resources available to the trial.

## Conduct problems in the classroom

Conduct problems in the classroom (primary outcome) were measured using the *Problem Behavior in the Classroom Last Week* scale [69]. The scale was translated from Norwegian to Swedish by the research group. It contains 20 items concerning challenging, disruptive or otherwise problematic behaviors that may occur in classrooms, such as verbal or physical assault. The teachers responded to a Likert scale from 0 (no observations last week) to 4 (several times a day)

regarding problem behaviors situated in the classroom during the previous week. The scale was rated by teachers at all timepoints (T1–T3). Previous Norwegian studies have reported an internal consistency of $\alpha = 0.86$ to 0.88 [69,70]. Internal consistency in our sample was estimated at $\omega_{within} = 0.91$ and $\omega_{between} = 0.93$.

### Conduct problems in common school areas

Conduct problems in common school areas were measured by the *Problem Behavior in the School Environment Last Week* scale [69]. The scale was translated from Norwegian to Swedish by the research group and consists of 15 items pertaining to challenging or problematic behaviors that may occur in common school areas (e.g., hallways, bathrooms, or schoolyards). Items are largely similar to the *Problem Behavior in the Classroom Last Week* scale. The teachers responded to a Likert scale from 0 (no observations last week) to 4 (several times a day) regarding problem behaviors in school areas (besides the classroom) observed during the previous week. The scale was rated by teachers at all time-points (T1–T3). A previous Norwegian study reported an internal consistency of $\alpha = 0.84$ [69]. Internal consistency in the present sample was estimated at $\omega_{within} = 0.85$ and $\omega_{between} = 0.97$.

### Teacher-rated classroom climate

The Norwegian version of the *Classroom Climate Scale* [69] was translated to Swedish by the research group and used to assess teacher-rated classroom climate. The scale consists of 14 items concerning several aspects of the classroom learning climate, such as task-orientation and engagement, relationships between students and with the teacher, and the absence or presence of conduct problems. Items were rated on a 0 (strongly disagree) to 3 (strongly agree). The scale was rated by teachers at T1 and T3. Previous Norwegian studies have reported an internal consistency of $\alpha = 0.83$ to 0.86 [69,70]. Internal consistency in our sample was estimated at $\omega_{within} = 0.65$ and $\omega_{between} = 0.93$.

### Observer-rated classroom climate

Observers rated classroom climate using a Swedish translation (translated by the research group) of the *Classroom Atmosphere Measure* [71]. Using this scale, the observer rates 10 items concerning different aspects of classroom climate: disruptive behavior, transitions, rule-following, cooperation, social problem-solving, expressing feelings appropriately, interest and engagement, focus and on-task behavior, responsiveness to students' individual differences, and support of students' efforts. Each item is rated on a scale of 1–4 using specific behavioral indicators, and 0 is used to indicate that the item was not possible to rate (e.g., if no transitions were observed). The original study measured inter-rater reliability using Cohen's kappa, which ranged from 0.62–0.81 for each item. Internal consistency was reported using Cronbach's alpha which was at $\alpha = 0.91$ [71]. Inter-rater reliability was not measured in the current sample due to time constraints during observer training. Internal consistency was estimated at $\omega_{within} = 0.69$ and $\omega_{between} = 0.79$ for this sample.

### On-task behavior

Observers assessed on-task behavior at the classroom level using a variation of momentary time sampling called Planned Activity Check [72]. At the end of each 3-min-interval the observers count the number of task-oriented students for a total of six intervals (18 minutes). The observers simultaneously kept track of the number of students by noting the total number of students before beginning the observation and then noting any incoming or departing students during the observation. Observers made 13 ratings during training, which were compared to a master-rater (DD) who had reviewed the video material before. Each observer's deviation from the master-rater was summarized using the median absolute deviation, which ranged from 0–1. An intraclass-correlation (ICC) was used to calculate inter-rater reliability for Planned Activity

Check during observer training; ICC = 0.78. There are different variants of ICC [73], this variant of ICC was chosen to take absolute agreement into account along with multiple observations per observer.

## GBG-certification

At the end of the school year the GBG implementation team certified teachers who they judged had reached the necessary criteria. The criteria for certification were attendance and participation in all three training sessions, having conducted at least 60 GBG-sessions, and whether teachers had passable levels of fidelity during GBG-sessions observed by a trainer.

Although tracking the number of sessions was part of the manualized instructions, it was not part of the GBG implementation team's usual practice at the time. Thus, the exact number of sessions is not known.

## Statistical analysis

Bayesian linear mixed models were used to estimate the difference in change over time between groups, primarily using interactions between time-points and group. Outcomes were aggregated, then standardized by subtracting the mean and dividing with the standard deviation. An exception was made for on-task behavior, which was modelled with a Bayesian binomial regression model using a logit link. Binomial regression can be used to model a count of successful trials, given the number of trials and probability of each trial being successful [74]. In this case, each rating of a student's on-task behavior can be considered a successful or unsuccessful trial, corresponding to if the student is on-task or not, and each observational interval is then a series of Bernoulli trials. All models adjusted for within-subject dependency induced by repeated measures by using random intercept for classroom ID. Possible dependency induced by stratification was adjusted for by using the stratification variable as a fixed effect [75]. As the number of schools were few, they were also added as fixed effects to further adjust for potential clustering [76].

Each Gaussian model had a corresponding model using Student's t-distribution or a skew normal distribution in case of better model fit due to outliers or skewness [77,78]. The final model was chosen based on parsimony, i.e., the least complex model. The more complex model was favored if it provided better fit to the data, which was defined as a difference in expected log pointwise predictive density approximated with leave-one-out cross-validation (LOOCV) being greater than 4, barring the difference being within 2 standard errors and any other issues with the model [79].

R (version 4.4.1) was the primary software used for statistical analysis, including visualizations. Bayesian model specification and diagnostics were facilitated by the R-package brms [80], which acts as a wrapper for the probabilistic programming language Stan [81]. Estimations and visualizations were facilitated by the R-packages emmeans [82], ggplot2 [83], and tidybayes [84]. Model sampling was performed with Markov Chain Monte Carlo, using four chains with 4000 iterations each. All final models converged well with estimates having Gelman-Rubin diagnostics (R̂) of 1.00, along with Tail- and Bulk Effective Sample Sizes of > 1000 [85].

Priors were chosen to be weakly informative, allowing estimates to be relatively unaffected with equal probability in each direction while also providing regularization and facilitating model convergence by lowering the probability of extremely unlikely estimates (e.g., effect sizes > ±2SD; [86]. The sensitivity for model choice was also explored using much wider priors, along with frequentist linear mixed effects models, and ordinal probit models, where the relevant estimates showed similar results to the final models. Relevant R-code for all analyses and diagnostics, including prior and posterior predictive checks, is publicly available together with the open data at Researchdata.se [87].

To assess differences in change over time between groups, results are presented as point estimates, which are the medians from the posteriors, along with the associated 90% highest posterior density interval (HDI), i.e., the 90% most likely values of the posterior [88]. A threshold of 66% and 90% for probabilities can be considered "likely" and "very likely", respectively [89]. Outcomes are also presented with the estimated posterior probability that the intervention group will have a larger change at the primary endpoint.

## Results

### Sample characteristics

Sample characteristics at baseline and primary endpoint are presented in Table 1. Most characteristics were similar between groups and over time. The overall sample's sociodemographic index is close to the municipalities assigned mean of 100, though the control group has a somewhat lower mean.

All teacher forms and observations were collected during the appointed time periods with a 100% response frequency and no missing data. Baseline data for the two classrooms that merged into adjoining classrooms were included in the analysis.

Raw unstandardized baseline values for the outcomes are summarized in Table 2 as means and standard deviations. Table 2 also contains robust Bayesian correlations (analogous to Pearson correlations) with corresponding 90% HDI.

**Table 1. Sample characteristics at baseline and primary endpoint.**

| Sample characteristics | GBG-group | | Control group | | Full sample | | |
|---|---|---|---|---|---|---|---|
| | T1 | T3 | T1 | T3 | T1 | T3 | Range |
| **No. of students** | | | | | | | |
| Total (n) | 611 | 617 | 377 | 373 | 988 | 990 | – |
| Mean per classroom | 23 | 25 | 24 | 23 | 23 | 24 | 17–28 |
| Mean number of male students per classroom | 12 (52%) | 13 (51%) | 12 (52%) | 12 (50%) | 12 (52%) | 12 (51%) | 8–17 (35%–71%) |
| Mean number of students with Swedish as secondary language per classroom | 8 (37%) | 8 (34%) | 9 (37%) | 8 (36%) | 8 (37%) | 8 (34%) | 0-24 (0%–100%) |
| Mean number of students wholly or partially excluded per classroom | 1 (3%) | 1 (5%) | 1 (5%) | 1 (4%) | 1 (4%) | 1 (4%) | 0–6 (0%–24%) |
| Mean sociodemographic index | 105 | – | 72 | – | 93 | – | 14–246 |
| **Additional staff present in classroom** | | | | | | | |
| No additional staff | 7 (26%) | 5 (20%) | 1 (6%) | 3 (19%) | 8 (19%) | 8 (20%) | – |
| Less than half the time | 11 (41%) | 5 (20%) | 6 (38%) | 7 (44%) | 17 (40%) | 12 (29%) | – |
| More than half the time | 9 (33%) | 12 (48%) | 6 (38%) | 4 (25%) | 15 (35%) | 16 (39%) | – |
| All the time | 0 (0%) | 3 (12%) | 3 (19%) | 2 (13%) | 3 (7%) | 5 (12%) | – |
| **Teacher experience with current students** | | | | | | | |
| New class | 15 (56%) | 0 (0%) | 6 (38%) | 0 (0%) | 21 (49%) | 0 (0%) | – |
| 1-2 terms | 5 (19%) | 13 (52%) | 4 (25%) | 6 (38%) | 9 (21%) | 19 (46%) | – |
| 3-4 terms | 4 (15%) | 7 (28%) | 5 (31%) | 7 (44%) | 9 (21%) | 14 (34%) | – |
| 5 or more terms | 3 (11%) | 5 (20%) | 1 (6%) | 3 (19%) | 4 (9%) | 8 (20%) | – |

Numbers are rounded to the nearest integer.

**Table 2. Baseline descriptive statistics and correlations for outcomes.**

| | Outcome | M | SD | 1 | 2 | 3 | 4 | 5 |
|---|---|---|---|---|---|---|---|---|
| 1 | Conduct problems in the classroom | 17.58 | 11.02 | — | 0.68 – 0.98 | −0.72 – −0.28 | −0.65 – 0.05 | −0.93 – −0.31 |
| 2 | Conduct problems in school areas | 9.77 | 8.82 | 0.83 | — | −0.53 – −0.09 | −0.40 – 0.19 | −0.50 – 0.09 |
| 3 | Teacher-rated classroom climate | 29.58 | 4.57 | −0.50 | −0.30 | — | −0.08 – 0.68 | 0.01 – 0.71 |
| 4 | Observer-rated classroom climate | 3.89 | 0.71 | −0.29 | −0.11 | .27 | — | 0.23 – 0.87 |
| 5 | On-task behavior[a] | 91% | 4% | −0.63 | −0.20 | .35 | .56 | — |

Table showing means (M) and standard deviations (SD) of unstandardized baseline values for each outcome. Estimated correlations on the bottom diagonal are posterior medians while the top diagonal values are the corresponding lower- and upper bounds of the 90% HDI.

[a]Values are derived from the mean proportion of on-task students during each observation at baseline.

Standardized scores are used to represent observed and model-based estimates for change throughout, with exception for on-task behavior. Table 3 summarizes model-based estimated change from baseline (T1) to primary endpoint (T3) for all outcomes by group. Estimates of change are presented using the posterior's median along with the corresponding lower- and upper bounds of the 90% HDI (in brackets). Each outcome is also visually summarized in figures, each showing individual and mean observations for each timepoint by group, model-based estimates represented by the posterior's median for each time point by group and associated 90% HDI, and model-based estimates of change between timepoints represented by the posterior's median and HDI.

By the end of the study, 20 of 25 remaining teachers in the intervention group were certified in GBG by the implementation team. Table 4 summarizes standardized means for teacher-rated outcomes in T1 and T3 by group, with the GBG-group divided into certified and non-certified. Observer-rated outcomes are not part of Table 4 as only one observer-rated classroom turned out non-certified.

## Conduct problems in the classroom

The observed and estimated change in conduct problems classrooms over time and by group is summarized in Fig 2. The estimated change in between T1 and T3 for the intervention group was −0.17 [−0.45, 0.11]. The corresponding change for the control group was −0.01 [−0.34, 0.30]. The biggest difference in estimated change between groups happens from T1 to T2, diminishing to a smaller difference in change between T2 and T3. The estimated posterior probability of the intervention group having a greater decrease in conduct problems in classrooms compared to the control group after a school year was 74%.

## Conduct problems in common school areas

The observed and estimated change in conduct problems in common school areas over time and by group is summarized in Fig 3. The estimated change in conduct problems in common school areas between T1 and T3 was 0.08 [−0.15, 0.33]

**Table 3. Summary of model-based estimates of change between T1 and T3.**

| Outcome | T3 − T1 [90% HDI] | |
|---|---|---|
| | GBG | Control |
| Conduct problems in the classroom | −0.17 [−0.45, 0.11] | −0.01 [−0.34, 0.30] |
| Conduct problems in school areas | 0.08 [−0.15, 0.33] | 0.06 [−0.20, 0.33] |
| Teacher-rated classroom climate | 0.22 [−0.06, 0.51] | −0.22 [−0.56, 0.15] |
| Observer-rated classroom climate | 0.55 [0.05, 1.04] | 0.07 [−0.48, 0.61] |
| On-task behavior[a] | 0.90 [0.70, 1.09] | 0.74 [0.52, 0.98] |

[a]Estimates of change for this outcome are represented by odds ratios.

**Table 4. Mean standardized outcomes by time and GBG-certification status.**

| Outcome | T1 | | | T3 | | |
|---|---|---|---|---|---|---|
| | GBG-C | GBG-NC | BAU | GBG-C | GBG-NC | BAU |
| Conduct problems in the classroom | 0.14 | 1.03 | −0.20 | 0.04 | 0.05 | −0.33 |
| Conduct problems in school areas | −0.12 | 1.29 | −0.32 | 0.04 | 0.17 | −0.24 |
| Teacher-rated classroom climate | −0.13 | −0.53 | 0.30 | 0.22 | −0.56 | 0.05 |

GBG-C, certified in GBG by June 2022; GBG-NC, not certified in GBG by June 2022; BAU, business-as-usual.

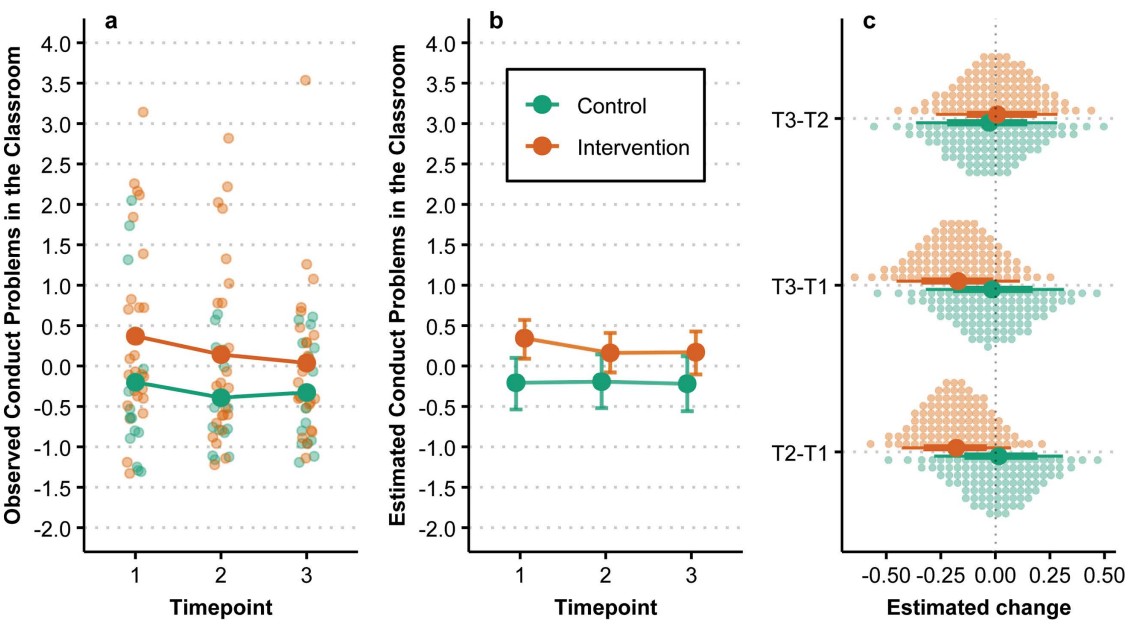

**Fig 2. Conduct problems in the classroom over time by group.** (a) Individual observations are denoted by smaller points while larger points denote observed group means. (b) Points are posterior medians and associated error bars represent the 90% HDI. (c) Large points are posterior medians, with associated lines representing the 66% and 90% HDI, while each small dot represents a percentile of the associated posterior.

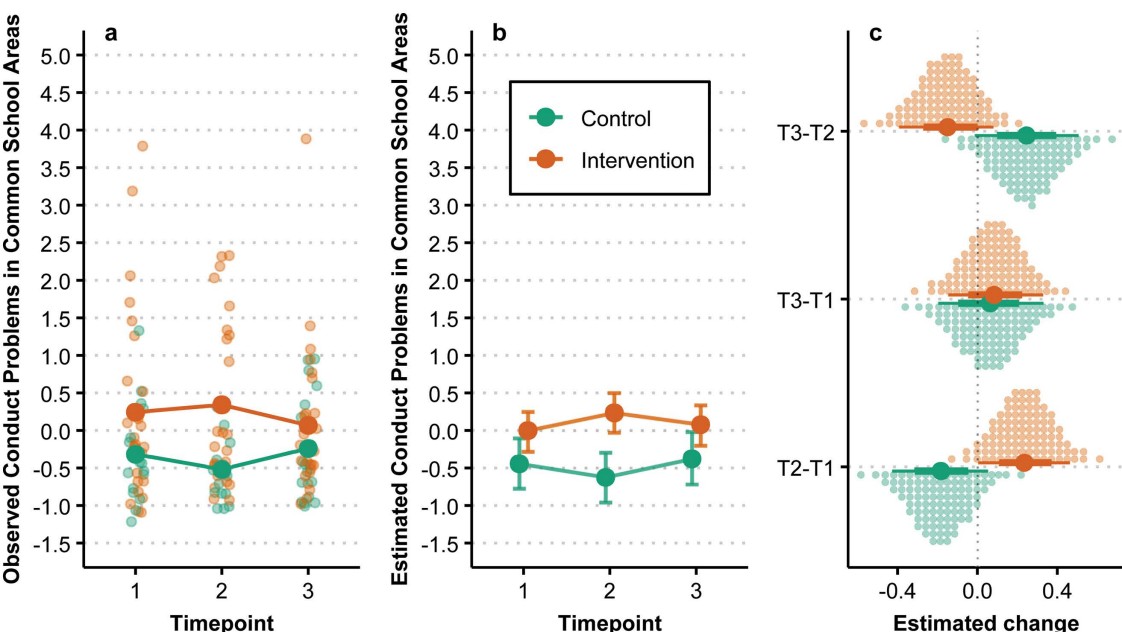

**Fig 3. Conduct problems in common school areas over time by group.** (a) Individual observations are denoted by smaller points while larger points denote observed group means. (b) Points are posterior medians and associated error bars represent the 90% HDI. (c) Large points are posterior medians, with associated lines representing the 66% and 90% HDI, while each small dot represents a percentile of the associated posterior.

for the intervention group and 0.06 [−0.20, 0.33] for the control group. Conduct problems in common school areas seem to increase in the intervention group and decrease in the control group from T1 to T2, before converging somewhat from T2 to T3. The estimated posterior probability of the intervention group having a larger decrease over time compared to the control group was 46%.

### Teacher-rated classroom climate

The observed and estimated change in teacher-rated classroom climate over time and by group, using standardized scores, is summarized in Fig 4. The estimated change in teacher-rated classroom climate between T1 and T3 was 0.22 [−0.06, 0.51] for the intervention group and −0.22 [−0.56, 0.15] for the control group. As seen in Fig 4, the 66% HDIs for estimated change by group do not overlap, while the 90% HDIs do overlap. As this outcome lacks a 3-month follow-up only the overall trend from baseline to primary endpoint is available, which shows an overall increase in teacher-observed classroom climate for the intervention group and an overall decrease in the control group. The estimated posterior probability of the intervention group having a greater increase over time compared to the control group was 94%.

### Observer-rated classroom climate

The observed and estimated change in observer-rated classroom climate over time and by group, using standardized scores, is summarized in Fig 5. The estimated change between T1 and T3 is 0.55 [0.05, 1.04] for the intervention group and 0.07 [−0.48, 0.61] for the control group. Overall, the estimates from T1-T3 show a steady increase for the intervention group compared to the control group. The estimated posterior probability of the intervention group having a greater increase over time compared to the control group was 88%.

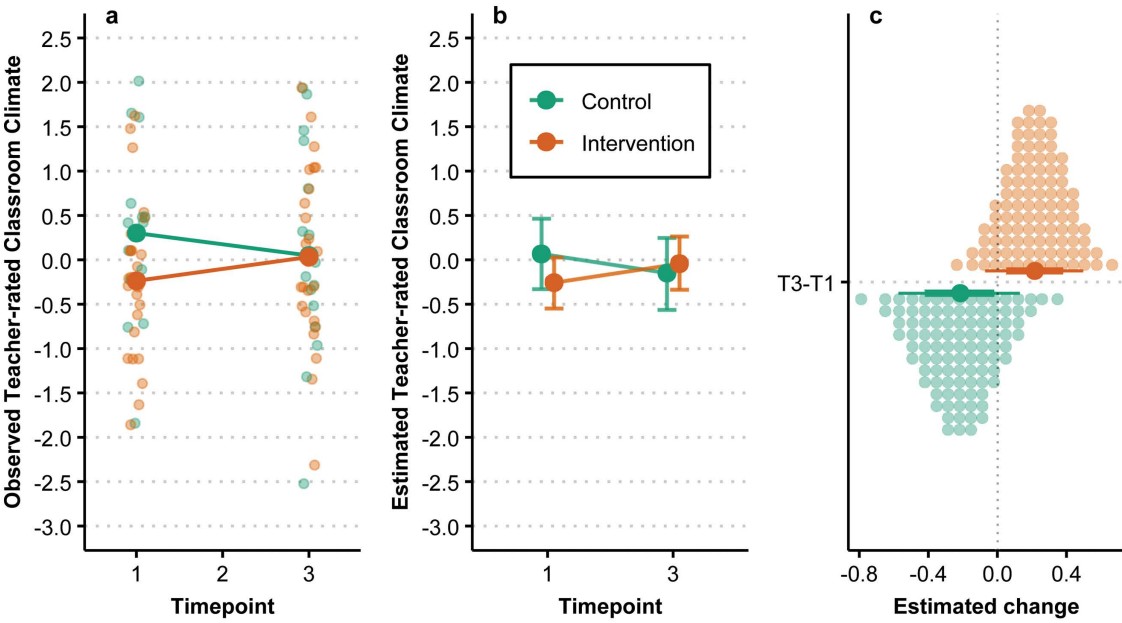

**Fig 4. Teacher-rated classroom climate over time by group.** (a) Individual observations are denoted by smaller points while larger points denote observed group means. (b) Points are posterior medians and associated error bars represent the 90% HDI. (c) Large points are posterior medians, with associated lines representing the 66% and 90% HDI, while each small dot represents a percentile of the associated posterior.

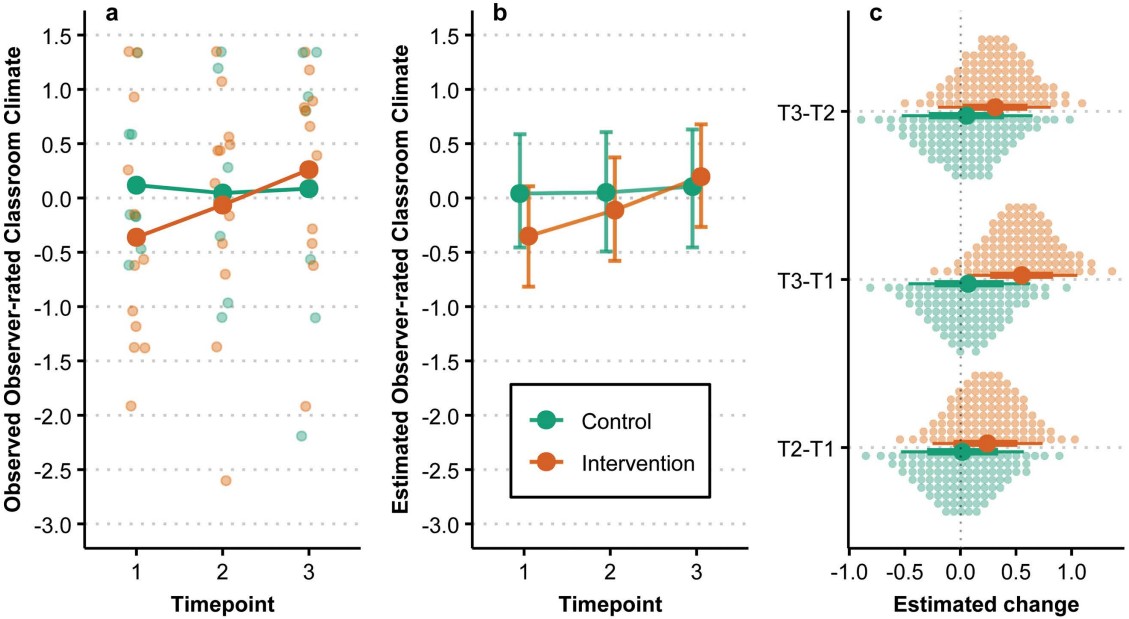

**Fig 5. Observer-rated classroom climate over time by group.** (a) Individual observations are denoted by smaller points while larger points denote observed group means. (b) Points are posterior medians and associated error bars represent the 90% HDI. (c) Large points are posterior medians, with associated lines representing the 66% and 90% HDI, while each small dot represents a percentile of the associated posterior.

### On-task behavior

The observed and estimated change in proportions of on-task behavior over time is summarized in Fig 6. The change over time is presented as an odds ratio (OR) between timepoints. As with other outcomes, the estimations are based on the posteriors' median and 90% HDI. The estimated relative change between T1 and T3 for the intervention group is OR = 0.90 [0.70, 1.09] and OR = 0.74 [0.52, 0.98] for the control group. Regarding absolute changes in probabilities, the intervention group has an estimated 0.92 [0.89, 0.95] probability of on-task behavior at T1 and 0.91 [0.88, 0.94] at T3, with the corresponding values being 0.94 [0.90, 0.96] at T1 and 0.91 [0.87, 0.95] at T3 for the control group. The estimated relative change between the T2 and T3 suggests that most of the relative increase in the intervention group takes place in this period. However, this is in relation to a relative decrease between T1 and T2. The estimated posterior probability of the intervention group having a greater OR for the change between T1 and T3 in on-task behavior compared to the control group was 80%.

### Discussion

This trial sought to evaluate the effectiveness of GBG-training over BAU in elementary school classrooms under pragmatic conditions. Barring one partial exception, differences in change over time are trending in the hypothesized directions, overall favoring GBG over BAU. However, findings are too inconclusive to support firm claims about specific outcomes, as the HDIs representing changes over time overlap. Given the imprecision, results warrant cautious interpretation grounded in previous research.

The primary objective of this trial was to evaluate the effects of GBG-training on conduct problems in the classroom. Hypothesis 1 stated that conduct problems in classrooms would have a greater decrease in the GBG-group compared to the BAU-group over time. The posterior estimates suggest that GBG decreases conduct problems in classrooms over time by a small measure, with most of the difference in change being prominent at phase one of GBG (i.e., between

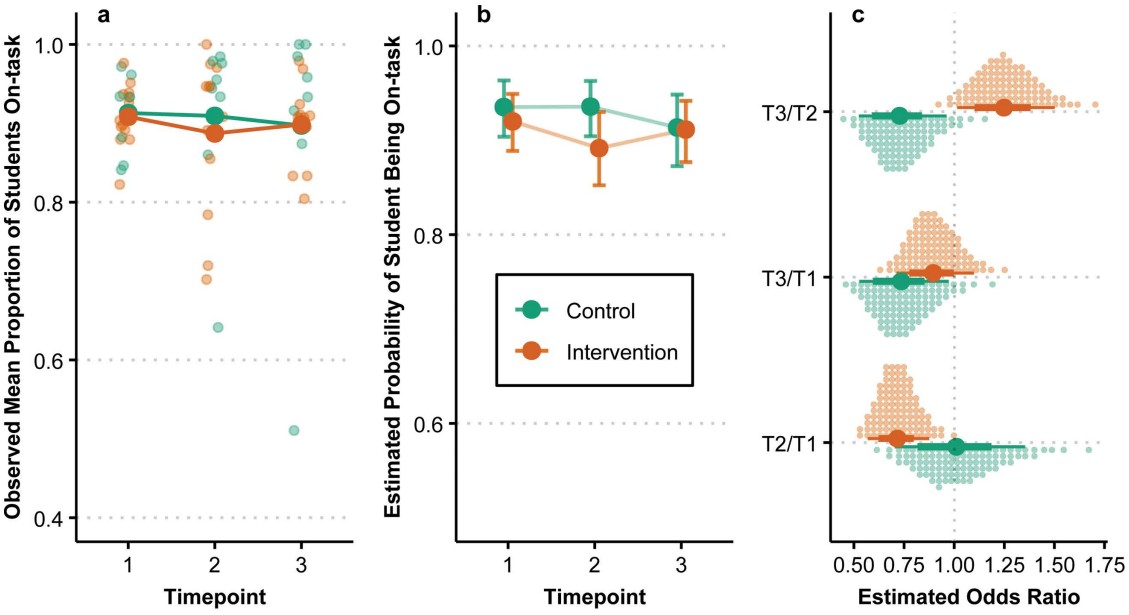

**Fig 6. On-task behavior over time by group.** (a) Small points denote the mean proportion of on-task students for each observed classroom at each timepoint, with larger points denoting group means. (b) Points are posterior medians representing the estimated probability of on-task behavior and associated error bars represent the 90% HDI. (c) Large points are posterior medians for relative change over time, with associated lines representing the 66% and 90% HDI, while each small dot represents a percentile of the associated posterior.

baseline and 3-month follow-up). Estimates are imprecise but in a positive direction and largely comparable to results from earlier studies regarding GBG and teacher-rated conduct problems [9]. The results are also in line with an earlier meta-analysis that examined school-based behavioral management programs and their effect on conduct problems in general [90]. Parallels can also be drawn to two Norwegian trials that evaluated translated and adapted versions of School-Wide Positive Behavior Support (SWPBS) and Incredible Years Teacher Classroom Management (IY-TCM). Both trials focused on school-based universal prevention using behavioral management and similar theoretical underpinnings as GBG, while also sharing a similar context and several measurement tools with this trial. Both trials showed similar effect sizes for conduct problems in the classroom, though effects were deemed non-significant in the IY-TCM trial [70,91]. The SWPBS trial also included an additional arm with a lighter and abridged version of SWPBS, which also showed improvements in this area [92]. The SWPBS trial had a longer follow-up time of three years with a slightly older age group, while the IY-TCM trial had a more similar timeframe and age group as compared to this trial. In addition to measuring conduct problems on a classroom level, the IY-ICM trial also studied changes in conduct problems on an individual level which resulted in standardized effect sizes around 0.08–0.09, with effects being more pronounced for children with higher baseline levels [93].

To summarize, although notably imprecise and modest, the reductions in teacher-rated conduct problems in the GBG-group compared to the control group are largely consistent with similar earlier research.

A secondary objective was to evaluate the effect of GBG-training on conduct problems in common school areas. Hypothesis 2 stated that the outcome was not expected to decrease at the 3-month follow-up compared to BAU, but rather at the 9-month follow up to match the increasing focus on generalization in GBG-components. Though imprecise, the estimates seem to follow the hypothesized direction as conduct problems in common school areas decrease from T2 to T3. However, this is preceded by an increase from T1 to T2 in the intervention group and a simultaneous decrease in the control group. The overall change over time between groups indicates a near nil effect on conduct problems in common school areas

within this timeframe. Although this outcome is highly correlated with conduct problems in the classroom at baseline, their overall trajectories by group are noticeably different. There is some limited research indicating that GBG is mostly effective in the specific context in which it is played [94,95]. While this version of GBG explicitly prescribes activities that are meant to promote generalization of behavior, these generalization activities are still conducted within the classroom context. Again, when comparing to the previously mentioned Norwegian studies, IY-TCM showed no positive effects on conduct problems in common school areas, while SWPBS did [70,91]. For SWPBS, this effect was already present after the first year of follow-up, and even the abridged version of SWPBS also showed notable improvements in the same outcome [69,92]. A notable difference from GBG is that SWPBS includes explicit strategies for reducing conduct problems in school areas besides the classroom. Consistent with existing research and intervention components, there is no evidence to indicate that this version of GBG has an effect on conduct problems in common school areas within a school year in a pragmatic setting.

We also aimed to evaluate the effect of GBG-training on classroom climate. Hypothesis 3 stated that both teacher-rated and observer-rated measures of classroom climate would have a larger increase in the GBG-group compared to the BAU-group over time. Although there is some lack of precision, both measures increase in the GBG-group relative to the control group over time. To our knowledge, there is little to no research examining the effect of GBG on general or aggregated constructs of the classroom's social and learning climate. Although the construct overlaps with some of the other measurements already in use (e.g., on-task behavior and presence or absence of conduct problems), there are some facets important to a healthy and supportive classroom climate that are not captured by the narrower measures, such as cohesion between students, relationships between students and teacher, and student engagement. Though it is worth mentioning that student relationships in the form of peer ratings generally improve with GBG [9,96]. SWPBS in Norway showed a positive effect on teacher-rated classroom climate [70]. Using the same measure, IY-TCM in Norway had an effect size of similar magnitude but with a high p-value, though a positive effect on student-teacher relationships specifically was found [91,97]. The abridged version of SWPBS in Norway didn't find effects on neither staff- nor student-rated classroom climate [92]. Taken together, there are notable effects from GBG on classroom climate, as rated by teachers and observers, albeit limited by some imprecision and mixed earlier research.

An additional objective was to investigate the effect of GBG-training on on-task behavior. Hypothesis 4 stated that on-task behavior would have a relative increase over time in the GBG-group compared to the BAU-group. Though estimates were imprecise in this case as well, both groups saw a small relative decrease in the odds of students being on-task over time, with the GBG-group having a somewhat smaller decrease, which is in the hypothesized direction. However, looking at the absolute effects, the groups only differed in a few percentage points in the probability of students being on-task over time, which begs the question of how practically significant the effect is. This sample seems to have comparatively high base rates for on-task behavior [98], which may affect results. The results are consistent with earlier research as previous randomized trials have shown that GBG increased on-task behavior as rated by observers compared to controls [99,100]. Additionally, GBG may have more impact on aggressive behavior when base rates of on-task behavior are low compared to peers [101].

Although there is too much imprecision to consider any single outcome highly probable, effects are by and large in line with previous research, which needs to be examined in the context and configuration of the current trial. A factor that should lend credibility to the findings is the use of teacher- and observer-rated measurements. While the teacher-rated and observer-rated measures of classroom climate are somewhat dissimilar regarding construct and methodology, results were quite similar. The different sources of measurement provide different viewpoints – teacher ratings provide a first-hand account of longer time periods, while observers provide a snapshot sample outside game sessions. An additional factor is that the independent and blinded observers were less subject to the same potential biases as teachers who were the primary intervention agents and aware of trial contingencies.

Another factor to consider is the results being largely consistent with the postulated mechanisms of GBG. As GBG prompts and reinforces behaviors within the classroom for the duration of the intervention, the bulk of the effects are

expected to take place in that context [94,95]. With successful generalization, effects are also expected to take place outside the game context, still within the classroom but eventually even outside the classroom. This is reflected in the results as the effects that are within the classroom and more closely related to GBG seem clearer and more robust (e.g., classroom climate) while effects pertaining to general self-regulation and other school environments are weaker and lagging. As previously stated, results are also broadly in line with previous research on GBG and other universal school-based behavioral management programs.

The Nordic context may indirectly factor into implementation and results. Sweden isn't necessarily extreme in any direction in regard to more demographic properties of education [47]. Instead, there may be more relevant factors to consider in a cultural sense. While there is a general recognition of behavioral issues as a concern, there isn't necessarily consensus on how to mitigate it. Attitudes may vary as the magnitude of the problem varies, and different schools or stakeholders may have different opinions of how to intervene. Programs based on applied behavior analysis have faced some resistance in Nordic countries, e.g., due to concerns regarding external motivation [45], or that group contingencies may evoke negative peer pressure or other forms of experienced punishment [102]. Although research on negative peer pressure and social validity generally is favorable for GBG [103–105], these controversies may constitute barriers that need to be addressed, whether they are misinformed or valid.

This trial was considered pragmatic, focusing on effectiveness and mirroring real-world conditions, rather than optimizing conditions to test efficacy. Consistent with the pragmatic focus, it is explicitly stated in this trial that neither intervention nor control group was given additional resources and that groups were fully flexible in terms of adherence. However, it is not clear to what degree this deviates as these factors are rarely elaborated on in earlier GBG-trials. Future research should be clear about trial objectives and conditions, and specify transparently according to guidelines, e.g., CONSORT 2025 [59]. Also in accordance with the pragmatic focus was that ITT was the mode of analysis rather than per-protocol. Both the less-than-ideal pragmatic conditions and ITT-analysis are expected to underestimate effects [106]. However, earlier research is not always straightforward on this matter. Efficacy trials usually have better intervention effects on conduct problems compared to effectiveness trials, but to some surprise, developer-led trials and trials in the original context are often outperformed by their counterparts (i.e., independent trials and trials in a new context) for conduct problems [40]. Although a direct conclusion isn't possible in this study, it is plausible that some aspects of naturalistic conditions could be advantageous, such as local adaptations by practitioners providing better contextual fit and efficiency when fidelity is less supervised. Potential reactive local adaptations are likely not captured by the standard fidelity forms used in this study.

The mode of analysis also means that implementation and related factors, such as adherence, dosage, and responsiveness, weren't in the scope of this paper. While there is an association between implementation and outcomes in a general sense for this type of intervention, there are considerable knowledge gaps concerning mechanics and specific facets of implementation [107]. For example, there is research suggesting that GBG could be efficacious despite low treatment integrity [108]. Future studies should continue to investigate these factors and how they relate to outcomes [e.g., 109–113], perhaps even advanced hybrid trials where different implementation strategies are evaluated [114]. Despite the study not being optimized for ideal conditions, 80% of the teachers receiving training in GBG were certified by the end of the school year, which is similar to previous studies [115]. Reasons for non-certification are unclear in this sample, though all teachers in the intervention group were exposed to GBG-training to some extent, and non-certified teachers could still be certified after the study period. More data from this study related to implementation and teacher practices will be analyzed and published separately as stated in the study protocol.

## Implications

Barring limitations, this study lends tentative support for the potential of universally preventative school-based interventions being transported cross-culturally amid highly pragmatic conditions while still providing improved outcomes. In this case, a small team of practitioners were able to initiate and complete the process of translating, adapting, building

capacity, and finally implementing an established intervention. This was likely facilitated by having the intervention provider somewhat close, culturally and geographically. It may still be pertinent to keep access to technical support from the intervention provider or otherwise have access to new research and ongoing developments, e.g., whole-day GBG [116], software-supported implementation [117,118], or adaptations to novel settings [119]. An implementation site may also need to take complex needs into account, e.g., for students with special needs. While GBG generally also provides gains for at-risk students, they may still require more intervention or adaptation than can be provided by universal prevention. This may require an implementation site to also have competence in special education, multi-tiered systems of support, or other ways of differentiating and adapting to special needs [32,120,121].

Though it is unclear how much effects differ in magnitude between versions of GBG, this study lends support to GBG being potentially efficacious despite adaptations. As previously mentioned, a notable adaptation is toning down the response-cost of rule-breaking behaviors. This is largely consistent with previous research as this adaptation has been successful in the earlier Dutch and Belgian studies using a similar version of GBG [100,122], and single-case trials showing only small differences [123,124]. Future research should be explicit when describing GBG as it changes and expands over time.

While findings from this study favor GBG being feasible in real-world conditions, it is not certain how conditions in this study generalizes to other contexts. For instance, while practitioners in this context operate on limited resources, they may still be able to offer greater implementation support compared to contexts with even less funding and resources. Conditions may also be affected by the implementation taking place within the context of a larger preventive framework (CTC) being piloted at the time. Training trainers and otherwise building capacity for an implementation team also requires initial and ongoing investments. It is also unclear if the whole-school approach to implementation is favorable compared to more classroom-specific implementation. The whole-school approach is likely to be more resource efficient, allowing for more simultaneous training and capacity building. At the same time, it may not be probable that all teachers in a school are predisposed to implementing GBG. As stated earlier, it is important for future research to be explicit and detailed regarding the conditions and context that are relevant to the trial and implementation.

Finally, practitioners looking to adopt GBG should be aware that both theory and empirical data suggest that any immediate effects are primarily situated in the classroom environment, rather than all school arenas. Gains in other environments may require additional efforts, interventions, or time.

## Limitations

The study's small number of clusters directly limits statistical power and ability to detect group differences, though stratified randomization helped mitigate potential imbalance. As it stands, imprecision is present both when interpreting estimates concerning the overall effect of GBG and estimates concerning the mechanisms (e.g., differences between specific timepoints). An alternative would be a more classroom-based randomization or recruitment, but this would mean risking contamination effects and a considerable change in how the municipality implements GBG, one of the pragmatic conditions this trial sought to mirror.

Connected to the sample size was the use of group-based measures, rather than individual. While group-based measures decrease available sample size and variability, it made the study less resource-intensive and possible to undertake in a highly pragmatic context, while maintaining a 100% response rate with no missing data. It can also potentially mitigate an issue within universal prevention, where many outcomes of interest have very low base rate values. Particularly in this case where the population largely consists of young and symptom-free children. This kind of non-individualized data is also easier to share with a high degree of availability.

Another limitation concerning measures was the predominant reliance on non-blinded teacher-rated measures due to the risk of potential biases. The independent and blinded observer-rated measures mitigated this issue somewhat, though these were limited by an even smaller sample size and a lack of on-site calibration.

## Conclusion

This randomized controlled trial sought to test the effectiveness of GBG under pragmatic conditions compared to BAU. Although estimates were imprecise due to sample constraints, findings were directionally consistent with prior evidence, lending cautious support to GBG's applicability under pragmatic conditions. Though the directional trends are not enough to constitute evidence of effectiveness, the study expands the literature on which settings, under what conditions, and with which adaptations GBG may be feasible.

## Supporting information

**S1 CONSORT Checklist. Checklist of information to include when reporting trial results.** CONSORT, Consolidated Standards of Reporting Trials. © 2025 Hopewell et al. This is an Open Access article distributed under the terms of the Creative Commons Attribution License (https://creativecommons.org/licenses/by/4.0/), which permits unrestricted use, distribution, and reproduction in any medium, provided the original work is properly cited.
(DOCX)

## Acknowledgments

The research group would like to thank and acknowledge all students, school personnel, the implementation team, and CTC-facilitators for making this trial possible.

## Author contributions

**Conceptualization:** Dariush Djamnezhad, Martin Bergström, Björn Hofvander.

**Data curation:** Dariush Djamnezhad.

**Formal analysis:** Dariush Djamnezhad, Martin Bergström, Carl Delfin, Björn Hofvander.

**Funding acquisition:** Dariush Djamnezhad, Martin Bergström, Björn Hofvander.

**Investigation:** Dariush Djamnezhad, Martin Bergström, Björn Hofvander.

**Methodology:** Dariush Djamnezhad, Martin Bergström, Carl Delfin, Björn Hofvander.

**Project administration:** Dariush Djamnezhad, Martin Bergström, Björn Hofvander.

**Resources:** Dariush Djamnezhad, Martin Bergström, Björn Hofvander.

**Software:** Dariush Djamnezhad, Carl Delfin.

**Supervision:** Björn Hofvander.

**Visualization:** Dariush Djamnezhad, Carl Delfin.

**Writing – original draft:** Dariush Djamnezhad.

**Writing – review & editing:** Dariush Djamnezhad, Martin Bergström, Carl Delfin, Björn Hofvander.

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
