## [Decision Letter · Decision Letter 0]

28 Oct 2025

PMEN-D-25-00453

Classroom effects of a preventive behavioral management program: A pragmatic cluster-randomized trial of Good Behavior Game

PLOS Mental Health

Dear Dr. Djamnezhad,

Thank you for submitting your manuscript to PLOS Mental Health. After careful consideration, we feel that it has merit but does not fully meet PLOS Mental Health’s publication criteria as it currently stands. Therefore, we invite you to submit a revised version of the manuscript that addresses the points raised during the review process.

We look forward to receiving your revised manuscript.

Kind regards,

Lambert Zixin Li, Ph.D.

Academic Editor

PLOS Mental Health

Journal Requirements:

1. Please send a completed 'Competing Interests' statement, including any COIs declared by your co-authors. If you have no competing interests to declare, please state "The authors have declared that no competing interests exist". Otherwise please declare all competing interests beginning with the statement "I have read the journal's policy and the authors of this manuscript have the following competing interests:"

1. Please clarify all sources of funding (financial or material support) for your study. List the grants (with grant number) or organizations (with url) that supported your study, including funding received from your institution. 

2. State the initials, alongside each funding source, of each author to receive each grant.

3. State what role the funders took in the study. If the funders had no role in your study, please state: “The funders had no role in study design, data collection and analysis, decision to publish, or preparation of the manuscript.”

4. If any authors received a salary from any of your funders, please state which authors and which funders.

Additional Editor Comments:

Thank you for submitting your manuscript to PLOS Mental Health. I invited four reviewers with relevant expertise, and most reviewers were enthusiastic about your study. I personally found your topic interesting and important, your methodology rigorous, and your findings potentially valuable for policy and practice.

However, the reviewers provided thoughtful feedback to further improve your paper. Please address all comments carefully, with particular attention to those raised by Reviewer 4.

We look forward to receiving your revised manuscript and detailed response memo.

Reviewers' comments:

Reviewer's Responses to Questions

**Comments to the Author**

1. Does this manuscript meet PLOS Mental Health’s publication criteria?

Reviewer #1: Yes

Reviewer #2: Yes

Reviewer #3: Yes

Reviewer #4: Yes

2. Has the statistical analysis been performed appropriately and rigorously?

Reviewer #1: Yes

Reviewer #2: Yes

Reviewer #3: Yes

Reviewer #4: Yes

3. Have the authors made all data underlying the findings in their manuscript fully available (please refer to the Data Availability Statement at the start of the manuscript PDF file)?

Reviewer #1: Yes

Reviewer #2: Yes

Reviewer #3: Yes

Reviewer #4: Yes

4. Is the manuscript presented in an intelligible fashion and written in standard English?

Reviewer #1: Yes

Reviewer #2: Yes

Reviewer #3: Yes

Reviewer #4: Yes

Reviewer #1: Article is well written. Accept.The study has been conducted well and manuscript is written flawlessly. Need for the study, methodology and results are understandable and scientific. Discussion and limitations are written adequately. Hence the manuscript can be considered for publication.

Reviewer #2: Based on the detailed review of your manuscript, I would recommend for the minor revision related to some parts of your paper:

1. Introduction part: Please condense history and focus on rationale because it is too lengthy and repetitive.

2. Challenges paragraph: Also merge ideas into one cohesive argument, check it because redundant phrasing

3. Discussion parts: Streamline and strengthen interpretive synthesis due to repetitive cautious tone.

4. Limitation parts: It is redundant sample size discussion, condense to one precise paragraph (“The study’s small number of clusters limits statistical power and generalizability, though stratified randomization helped mitigate potential imbalance”).

5. Conclusion: Please replace with more scholarly terms ("imprecise", "low precision")

"As effects were relatively uncertain, evidence on specific outcomes and their trajectories are inconclusive...") => "Although estimates were imprecise due to sample constraints, findings were directionally consistent with prior evidence, lending cautious support to GBG’s applicability under pragmatic conditions”.

6. add one concise summary table of key Bayesian results for clarity, that would be helpful

7. Shorten figure captions that currently restate results in full sentences (simplify and align with journal style).

Reviewer #3: please see the attached document for comments. also,the manuscript needs to be language edited and formatted making sure that it is written in the past tense and not future tense as it is a report from a completed study.

Reviewer #4: The topic is highly relevant to school mental health, universal preventive interventions, and implementation science. The manuscript is well-organized, transparent in methods, and written clearly.

However, several issues limit interpretability and the strength of conclusions.

1. The study is underpowered due to having only five randomized clusters (schools). While the authors acknowledge this, it needs stronger, clearer articulation:

a. Emphasize that the small number of clusters directly limits inferential certainty and ability to detect group differences.

b. Explicitly state that directional trends should not be interpreted as evidence of effectiveness, only feasibility and plausibility.

c. Consider adding a simulation-based or literature-based justification of the detectable effect size under the current design.

2. The manuscript states this was a “highly pragmatic” trial, but does not clearly describe:

a. How pragmatic decisions influenced implementation fidelity and support.

b. Which components differed from “standard GBG efficacy implementation models.

3. The manuscript frames results as “tentative support.” This is reasonable, but certain phrasing throughout reads as overly optimistic given overlapping HPD intervals (e.g., “Effects were largely in a favorable direction” and “This study lends tentative support for cross-cultural transportation”). I recommend strengthening the language distinguishing statistical uncertainty vs. practical plausibility, especially in the Discussion and Conclusion.

4. Teacher-rated conduct problems and climate are not blinded, and intervention teachers were primed through training. This creates risk of expectancy and reporting bias. This must be acknowledged more centrally as a study limitation, particularly because observer outcomes carry “n = 20 classrooms” and minimal rater–calibration data.

5. The paper notes that 80% of teachers were certified, but does not analyze outcomes by certification status. Even a descriptive comparison would strengthen interpretability.

6. Phrases like “effects were uncertain” and “consistent with earlier research” in the Abstract are slightly contradictory. Please clarify that consistency refers to direction, not magnitude or certainty.

**Do you want your identity to be public for this peer review?** For information about this choice, including consent withdrawal, please see our Privacy Policy

Reviewer #1: No

Reviewer #2: **Yes: ** Samphoas Chien

Reviewer #3: **Yes: ** Dr. Nkarenbi Juliette Bih

Reviewer #4: No

---

## [Editor Report · Decision Letter 1]

11 Dec 2025

Classroom effects of a preventive behavioral management program: A pragmatic cluster-randomized trial of Good Behavior Game

PMEN-D-25-00453R1

Dear Mr Djamnezhad,

We are pleased to inform you that your manuscript 'Classroom effects of a preventive behavioral management program: A pragmatic cluster-randomized trial of Good Behavior Game' has been provisionally accepted for publication in PLOS Mental Health.

Best regards,

Lambert Zixin Li, Ph.D.

Academic Editor

PLOS Mental Health

Dear Authors,

Thank you for your revision. The manuscript now meets the journal’s standards, and I am pleased to accept it for publication. I especially commend the paper for its novelty and practical implications. Congratulations on your excellent work.

Sincerely,

Lambert Zixin Li, PhD